# Standardizing an Experimental Murine Model of Extraparenchymal Neurocysticercosis That Immunologically Resembles Human Infection

**DOI:** 10.3390/brainsci13071021

**Published:** 2023-07-01

**Authors:** Alejandro Espinosa-Cerón, Alejandro Méndez, Juan Hernández-Aceves, Juan C. Juárez-González, Nelly Villalobos, Marisela Hernández, Georgina Díaz, Paola Soto, Luis Concha, Iván N. Pérez-Osorio, Juan J. Ortiz-Retana, Raúl J. Bobes, Robert M. Parkhouse, P. T. Hamamoto Filho, Gladis Fragoso, Edda Sciutto

**Affiliations:** 1Instituto de Investigaciones Biomédicas, Universidad Nacional Autónoma de México, Ciudad de México 04510, Mexicogladis@unam.mx (G.F.); 2Facultad de Medicina Veterinaria y Zootecnia, Universidad Nacional Autónoma de México, Ciudad de México 04510, Mexico; 3Instituto de Neurobiología, Universidad Nacional Autónoma de México, Querétaro 76230, Mexico; 4Instituto Gulbekian de Ciência, Portugal. R. Q.ta Grande 6, 2780-156 Oeiras, Portugal; 5Department of Neurology, Psychology and Psychiatry, Botucatu Medical School, UNESP-Universidade Estadual Paulista, São Paulo 18618-687, Brazil

**Keywords:** neurocysticercosis, *T. crassiceps*, experimental model, neuroinflammation

## Abstract

Background: Neurocysticercosis (NCC) is endemic in non-developed regions of the world. Two forms of NCC have been described, for which neurological morbidity depends on the location of the lesion, which can be either within the cerebral parenchyma or in extraparenchymal spaces. The extraparenchymal form (EXP-NCC) is considered the most severe form of NCC. EXP-NCC often requires several cycles of cysticidal treatment and the concomitant use of glucocorticoids to prevent increased inflammation, which could lead to intracranial hypertension and, in rare cases, to death. Thus, the improvement of EXP-NCC treatment is greatly needed. Methods: An experimental murine model of EXP-NCC, as an adequate model to evaluate new therapeutic approaches, and the parameters that support it are described. EXP-NCC was established by injecting 30 *Taenia crassiceps* cysticerci, which are less than 0.5 mm in diameter, into the cisterna magna of male and female Wistar rats. Results: Cyst implantation and infection progression were monitored by detecting the HP10 antigen and anti-cysticercal antibodies in the serum and cerebral spinal fluid (CSF) of infected rats and by magnetic resonance imaging. Higher HP10 levels were observed in CSF than in the sera, as in the case of human EXP-NCC. Low cell recruitment levels were observed surrounding established cysticerci in histological analysis, with a modest increase in GFAP and Iba1 expression in the parenchyma of female animals. Low cellularity in CSF and low levels of C-reactive protein are consistent with a weak inflammatory response to this infection. After 150 days of infection, EXP-NCC is accompanied by reduced levels of mononuclear cell proliferation, resembling the human disease. EXP-NCC does not affect the behavior or general status of the rats. Conclusions: This model will allow the evaluation of new approaches to control neuroinflammation and immunomodulatory treatments to restore and improve the specific anti-cysticercal immunity in EXP-NCC.

## 1. Introduction

Cysticercosis, which is caused by the cystic larvae of the cestode *Taenia solium,* is an endemic disease in most countries of Latin America, sub-Saharan Africa, and Asia [1,2]. The main causes for the persistence of this parasitosis are a lack of access to clean water and drainage, poor education, free-roaming pigs with access to contaminated human feces, and a lack of control in pig trade and consumption [2]. The life cycle of the parasite includes an adult stage, the tapeworm that inhabits the human small intestine. This intestinal infection can be acquired by the ingestion of poorly cooked pork meat infected with cysticerci. Pigs become infected when they ingest eggs produced by this tapeworm in human feces. Humans can also acquire cysticercosis by accidentally ingesting tapeworm eggs in contaminated food and/or water [2].

The most serious form of the disease occurs when the parasite lodges in the central nervous system. Neurocysticercosis (NCC) is a foodborne neglected parasitic disease that causes 2.8 million disability-adjusted life years (DALYs) due to seizures, epilepsy, and intracranial hypertension [3]. Two main forms of NCC have been described [4]. Parenchymal neurocysticercosis (P-NCC) occurs when the parasite is located in the brain parenchyma and is generally benign. In P-NCC, the parasite(s) usually die without any treatment. Most P-NCC cases do not cause symptoms requiring consultation. Indeed, lesions compatible with NCC were found in approximately 9% of participants in various neuroepidemiological studies conducted in rural communities of Mexico (States of Puebla and Morelos) [5,6]. Despite its usually benign evolution, clinical studies have shown that cysticidal treatment improves the evolution of P-NCC [7,8,9]. The other form of NCC, known as extraparenchymal neurocysticercosis (EXP-NCC), occurs when the parasite is located outside the parenchyma, frequently in the ventricles or in the subarachnoid space at the basal cisterns [10]. The proportion of patients with P- or EXP-NCC may change depending on exposure-related factors and differences in genetic, immunologic, and sexual factors between populations. In Mexico, it is estimated that more than half of patients who attend referral centers such as the National Institute of Neurology and Neurosurgery suffer from EXP-NCC [4]. In these locations, the parasite can remain without causing symptoms for a long time [11], making diagnosis difficult, although magnetic resonance imaging (MRI) can help assess the number, size, and location of parasites [12]. Eventually, due to the longer residence time of the parasite, its detection by the host immune system, or some concomitant inflammatory condition, an exacerbated inflammatory process occurs, inducing intracranial hypertension and a depressed peripheral cellular immune response [13], with focal neurological deficits (16%) or recurrent seizures occurring in about 80% of symptomatic cases [14]. Some degree of cognitive dysfunction has been reported in up to 88% of NCC patients, as well as significant deficits in motor control and impulsivity [14]. Ventriculostomy is often performed to reduce intracranial hypertension in cases of hydrocephalus. EXP-NCC requires cysticidal treatment co-administered with high levels of glucocorticoids to control neuroinflammation and prevent intracranial hypertension, which can be life-threatening [15]. Administering albendazole at 30 mg/kg/day for 10 days, along with dexamethasone at 0.4 mg/kg/day for 13 days, followed by a tapering with prednisone, is the most common schedule [16]. Unfortunately, only less than 30% of patients show a total response to this treatment. Non-responders require repeated cycles of treatment [17]. Moreover, a non-negligible percentage of patients fail to respond to repeated courses of medication. What is worse, parasites increase in size in some patients during treatment, and new parasites may even appear during follow-up [16]. This scenario points to the importance of improving the treatment currently available.

When optimizing the treatment for EXP-NCC, it is crucial to consider the effect of glucocorticoids. Whilst required to control the exacerbated inflammation that puts the life of patients at risk, glucocorticoids are known to inhibit the specific immunity against the parasite [18]. This effect is not irrelevant to the efficiency of the response to cysticidal treatment. Indeed, our studies indicate a possible participation of proinflammatory cytokines in an effective response to cysticidal drugs [19]. Thus, more effective administration routes for glucocorticoids, like the intranasal route [20], or the use of other non-steroidal anti-inflammatory drugs have been proposed to improve the response to cysticidal drugs. To assess these alternative approaches, NCC models that accurately resemble human EXP-NCC are required.

Several NCC models have been developed in rodents and pigs [21,22,23]; however, they have been mainly used to describe the pathogenic effects of the presence of the parasite or the immune response induced by the infection.

A rat model of EXP-NCC previously described by Hamamoto-Filho et al. (2015) [24] was reproduced with some modifications and is implemented in our study with the aim of using it to evaluate new therapeutic approaches to restore the specific immunity against the parasite, to evaluate new cysticidal drugs and more effective interventions, and to reduce neuroinflammation.

## 2. Material and Methods

### 2.1. Animals

Groups of 7–10 female and/or male Wistar rats (*Rattus norvergicus*) aged 8–9 weeks were assigned to the treatments described below following a random variable simulation method implemented in R. Groups of 4–10 non-infected (naïve) or sham female or male rats were also included as controls in each experiment. All rats were maintained at 22 ± 3 °C under a 12/12 h light–dark cycle and with free access to water and food. Each rat was ear-tagged with a code with a consecutive number. All animals were weighed before experimental infection and weighed 30 and 60 days after. The weights of 9-week-old naïve rats were recorded as a baseline. All housing and experimental procedures were approved by the Institutional Committee for the Use and Care of Laboratory Animals of the Instituto de Investigaciones Biomédicas, UNAM, permit number ID 6313. Considering the stable conditions of the animals, no inclusion or exclusion criteria were established at the beginning of the study. Every effort was made, especially during surgery and euthanasia, to minimize animal suffering and stress.

### 2.2. Taenia crassiceps Cysts and Inoculation

A previously described procedure [24] was used herein, with minor modifications to the number of parasites and the method for cysts injection. Briefly, the parasites for inoculation were obtained from the peritoneal cavity of mice previously infected with *Taenia crassiceps* cysts, ORF strain (less than 3 months of infection). Upon aseptic removal, cysticerci were washed three times with sterile phosphate-buffered saline (PBS) and kept at 37 °C. Cysticerci measuring 400–500 μm in diameter were selected under a stereoscopic microscope. Then, 30 cysts were placed in a 20-cm-long 0.010” ID × 0.030” OD Tygon^®^ Masterflex microbore transfer tubing (Cole-Parmer, Vernon Hills, IL) containing 50 μL of PBS, connected to a 1 mL syringe at one end and a 25 G needle at the other.

Rats were first anesthetized with 5% isoflurane in oxygen for 5 min and maintained with 3% isoflurane during the inoculation procedure. The anesthetized animals were secured in a stereotactic frame, maintaining the head at an 80–90° angle to identify the depressible surface between the occipital protuberance and the atlas spine. The cerebromedullary cistern was punctured, and the rats were inoculated with the 30 metacestodes previously loaded into transfer tubes. A group of 4 sham-operated animals underwent the same procedure, but they received cyst-free PBS. Serum samples were obtained by retro-orbital bleeding, following the NIH Office of Animal Care and Use guidelines [25], at different post-infection times, as shown in Figure 1 and Figure 2. Cerebrospinal fluid (CSF) samples (100–150 μL) were obtained at different times post-infection via a puncture at the inoculation site with extraction probes, using transfer tubing of the same gauge, before the animals were euthanized.

### 2.3. HP10 Ag-ELISA Assay

HP10 antigen detection, which has been reported to be valuable to identify viable cysticerci, was used to monitor infection, following a previously described Ag-ELISA assay [26] with minor modifications. Both CSF and serum samples were assayed in duplicate. Briefly, the plates (Nunc, Rochester, New York, NY, USA) were coated with monoclonal anti-HP10 antibody (MoAb) diluted to 1 µg/100 µL per well in borate-buffered saline (BBS), pH 8.2, and incubated overnight at 4 °C. The plate wells were blocked with bovine serum albumin (Roche, Ciudad de México, México) in PBS (1.0% *w*/*v* and 0.05% *v*/*v* Tween 20) and left for 60 min at room temperature. Between steps, the plates were washed four times with 0.15 M NaC1 (*w*/*v*)/0.02% (*v*/*v*) Tween 20 in a Thermo Scientific Wellwash (Waltham, MA, USA). Undiluted samples (85 µL/well) were added and incubated for 60 min at 37 °C. Biotinylated anti-HP10 MoAb diluted 1:1000 and horseradish-peroxidase-conjugated streptavidin (Zymed, San Francisco, CA, USA) diluted 1:4000 were added consecutively, and the plates were incubated for 45 min at 37 °C. Then, tetramethylbenzidine (Invitrogen Carlsbad, CA, USA) was added as substrate. The reaction proceeded for 30 min at room temperature in the dark, and it was stopped by adding 100 µL of 0.2 M H_2_SO_4_ (Baker, Estado de Mexico, Mexico). Optical density (OD) was read at 450 nm in an ELISA processor Opsys MR Dynex Technology (Chantilly, VA, USA).

### 2.4. Antibody Detection

To monitor the humoral response induced by the infection, serum levels of anti-cysticercal IgG antibodies were measured by indirect ELISA, using *T. crassiceps* cyst fluid as antigen source, following a previously described procedure with minor modifications [27]. After incubation with 100 μL of *T. crassiceps* cyst fluid (1 μg/mL) in BBS, pH 9.6, for 60 min at 37 °C, the wells were washed and incubated with 100 μL of rat serum diluted 1:100 in PBS-Tween for 60 min at 37 °C. Antibodies were detected with 100 μL of alkaline phosphatase conjugate (whole molecule) rabbit anti-rat IgG (Sigma) and diluted to the optimum concentration (1:1000), followed by the addition of the substrate (para-nitrophenylphosphate, Sigma, 5 mg/mL). The reaction was stopped using 50 μL/well of 2N NaOH, and OD was read at 405 nm in an ELISA reader (Human GMBH, Humareader ModeI 2106, Wiesbaden, Germany). A serum sample was considered positive when OD reads exceeded the 99% confidence interval (mean value of the (non-immunized) control +3 standard deviations).

### 2.5. Magnetic Resonance Imaging (MRI)

In vivo MRI was performed on days 30 and 120 post-infection to detect the presence of extraparenchymal cysticerci. The animals were anesthetized with isoflurane and maintained at room temperature. A 7 T Bruker BioSpec 70/16 scanner (Bruker Biospin, Billerica, MA) was used. Images were acquired using a 2D T2 sequence with echo time/repetition time = 3.5/223 ms, three averages, echo train length = 1, flip angle of 60°, and slice thickness of 0.5 mm. The software RadiAnt DICOM Viewer was used for image analysis.

### 2.6. Histological and Immunofluorescence Analysis

To describe the stage of cysticerci and the inflammatory reaction induced by their presence in brain tissues, a histological analysis was performed. The animals were euthanized 120 days post-infection. A cardiac perfusion was performed with saline solution on anesthetized rats, followed by 10% buffered formalin for brain fixation. Brains were removed, carefully checking the meninges and the proximal regions to the cisterna magna under a stereoscopic microscope for the presence of parasites. For histological analysis, 4 µm thick coronal sections of the brain were obtained every 500 μm from the optic chiasm to the terminal region of the cerebellum; then, these sections were prepared and stained with hematoxylin–eosin. For immunohistological studies, frozen brains were cut into 30 µm thick coronal sections. Samples were washed thoroughly with Tris-buffered saline (TBS) and blocked with a 2% albumin solution (IgG-free albumin; Sigma, Saint Louis, MO, USA) in TBS for 30 min at room temperature. Brain sections were incubated overnight at 4 °C with goat anti-GFAP pAb (Abcam, Cambridge, UK) and rabbit anti-Iba-1 (Wako Chemicals, Inc., Richmond, VA, USA) in TBS-2% BSA to detect astrocytes and microglia, respectively. After washing, the sections were incubated for 1 h at room temperature with AlexaFluor 488 donkey anti-goat IgG (Abcam, Cambridge, UK), and AlexaFluor 594 goat anti-rabbit IgG (Molecular Probes, Eugene, OR, USA) diluted in TBS-2% BSA. The samples were mounted with a Vectashield medium (Vector Laboratories, Burlingame, CA, USA) containing 4′,6-diamidino-2 phenylindole (DAPI) for nuclei imaging. Photographs were obtained using a digital camera attached to a light microscope (Nikon Digital Sight DS-Ri1, Nikon, Tokyo, Japan). Regions close to the infection sites that were observed by MRI were selected and processed using the software ImageJ. The same software was employed to estimate fluorescence intensity due to Iba-1 or GFAP, as well as the number of positive cells for each of these markers in the regions of interest.

### 2.7. Peripheral Blood Mononuclear Cell Isolation

The rats were anesthetized with isoflurane (5%), and blood samples were collected by retro-orbital bleeding. The blood was heparinized with 25 U.I./2 mL of blood. PBMCs were resolved with Lymphoprep^®^ StemCell (1.077 g/mL, Cat. 07851), following the manufacturer’s directions to obtain mononuclear cells. When the sample contained erythrocytes, the cells were incubated with red blood lysis buffer for 5 min and washed. Cell viability was evaluated by the Trypan blue exclusion method, and only viable cells were considered for functional assays.

### 2.8. Proliferation Assay

Considering that human EXP-NCC is usually accompanied by a depressed lymphocyte proliferative capacity, this parameter was studied in our rat model. PBMCs isolated from all rats were stained with 5 µM CFSE (Biolegend, San Diego, CA, USA) diluted in RPMI-1640 for 10 min. The cells were washed with RPMI-1640 supplemented with 10% fetal bovine serum. Then, 10^5^ cells were cultured in a 98-well plate (Corning, New York, NY, USA) at 37 °C under 5% CO_2_. To check for proliferation potential, the cells were stimulated with either 1 µg/mL concanavalin A or medium for 96 h. Thereupon, the cells were recovered and analyzed by flow cytometry. Cells with low complexity and size were selected for proliferation analysis. Data was acquired on a NexT Attune cytometer with two lasers (red and blue) and analyzed with the software FlowJo X 10.0.7r2.

### 2.9. C-Reactive Protein Measurement (CRP)

As CRP is a classic inflammation indicator, serum CRP levels were measured with an ELISA Abcam commercial kit (ab256398), following the manufacturer’s directions.

### 2.10. Behavioral Tests

The open field test (OFT) was used to determine whether the infection modified the general locomotive activity of rodents. OFT had been previously used to assess the behavior of rats in this model [28]. The OFT apparatus consisted of a 100 cm × 100 cm × 45 cm box built of PVC sheets. Each rat was maintained in the box for 5 min, and its movements were videotaped from above with a video camera framing the entire surface of the box. The field was divided into sixteen 18 cm × 18 cm squares by superimposing a grid onto the video. The recordings were analyzed by independent (blinded) observers. The parameters assessed were horizontal movements (number of lines crossed), vertical movement or rearing (number of times in hind-limb support), and time in the central square (length of time spent in the center square), as described previously [28].

### 2.11. Statistical Analysis

The infection was monitored from day 0 to day 120 post-infection. On day 120 post-infection, the presence of cysticerci was confirmed by direct observation, microscopic analysis, HP10 detection, and MRI. No signs of distress or locomotor abnormalities were observed. The effect of different treatments was assessed at 4 months post-infection, before any damage associated with the presence of the parasite appeared.

The number of rats per group was estimated based on the results of Hamamoto-Filho et al. [28], considering the higher efficacy of infection in our study (60% vs. >90%). Data normality was tested with the Shapiro–Wilk test. Differences between mean OD, behavioral data (vertical and horizontal movements), body weight loss, and fluorescence intensity in immunofluorescence assays were evaluated by one-way ANOVA, followed by Dunnett or Tukey test. A two-tailed Student’s *t*-test was used to compare proliferation capacity in PBMCs. Behavioral data in the central square were evaluated with a Kruskal–Wallis test. Differences were considered as statistically significant when *p*-value was less than 0.05 *, 0.01 **, or 0.0001 ***. All analyses were carried out with GraphPad Prism^®^ v.8.0 (GraphPad Software, San Diego, CA, USA) and R v.4.30.

## 3. Results

### 3.1. Establishment of Extraparenchymal Neurocysticercosis and Follow-Up by HP10 Detection

In a pilot study, NCC was induced in 7 rats (4 females and 3 males) by direct implantation in the cerebromedullary cistern. The HP10 antigen was measured every 10 days for the first 60 days post-infection. HP10 levels showed a progressive increase after inoculation, which became significant on day 40 post-infection (*p* < 0.01). HP10 levels decreased after day 40 post-infection only in one male (Figure 1A). Serum levels of anti-*T. crassiceps* antibodies were significantly increased after day 40 post-infection (*p* < 0.0001) (Figure 1B).

Less weight gain was observed in the EXP-NCC rats compared to the control rats (non-infected rats, Ctr), which was more marked in females (*p* = 0.41) than in males (*p* = 0.60) (Figure 1C,D). In addition, the size of cysticerci in CSF samples obtained directly from the cisterna magna on day 60 post-infection (Figure 1F) was clearly larger than that of the parasites when inoculated at the onset of infection (Figure 1E).

In this study, no inflammatory cells were found in the CSF of infected rats. The levels of HP10 in sera were compared with those in CSF in a group of 20 rats (10 females and 10 males), which were followed up to day 120 post-infection. As previously noted, animals with cysticerci showed a time-dependent increase in the levels of HP10 antigen, with higher values in CSF samples from day 10 post-infection (Figure 2). HP10 levels did not increase after infection in 4 of these 20 rats.

### 3.2. Detecting Extraparenchymal Cysticerci by MRI

After cyst inoculation in a pilot experiment, the presence and distribution of cysts in the brain of rats were verified by MRI. Infected animals exhibited a time-dependent increase in parasite size. Cysticerci were not visible yet by MRI in one rat on day 30 post-infection (Figure 3A). However, an image compatible with a cysticercus was found in the periphery of the cisterna magna of the same rat on day 120 post-infection (Figure 3B). The other four animals analyzed showed images compatible with cysts in the basal cisterns, the cisterna magna, or in the spinal subarachnoid space. No intraparenchymal parasites were observed (Figure 3C–F). Racemose cysticerci were found on postmortem examination in the brain regions of interest on day 120 post-infection (Figure 3G,H). No clear evidence of hydrocephalus or ventricle enlargement was observed.

### 3.3. Histological Analysis

The seven rats in the pilot experiment were euthanized on day 120 post-infection. Upon brain removal, a thorough search for parasites was performed from the optic chiasm to the caudal medulla oblongata near the caudal termination of the cerebellum. A histological analysis was performed on H&E-stained, paraffin-embedded sections from four of seven rats. Cysticerci were found in three of the four animals studied. The parasites were located near the cisterna magna (between the cerebellum and the atlanto-occipital membrane), in the extraparenchymal/meningocortical region, or in the meninges near the brain stem. No anomalies or remarkable histological changes were observed in healthy rats, except for slight congestion (Figure 4A). No obvious pathological changes were observed in the animals studied, even when cysticerci were found at the postmortem examination. A mild inflammatory reaction was observed, along with small hemorrhagic areas and edema, which was unrelated to the presence of parasites (Figure 4C–E). In this study, no increase in serum CRP levels was observed on day 180 post-infection.

### 3.4. EXP-NCC Is Accompanied by Astrocyte activation Only in Female Rats

Since neurocysticercosis may trigger an inflammatory response, the expression of Iba-1 and GFAP (two classic markers of inflammation in the brain) was measured on day 120 post-infection. HP10 levels remained low in one of the rats in the pilot experiment and no parasites were found on postmortem examination; thus, this animal was not included in the immunofluorescence analysis. Iba-1 and GFAP expression was assessed in the two remaining animals harboring parasites, as well as in a healthy rat (Figure 5A). A higher GFAP expression was observed in the infected female with respect to the control (*p* < 0.05) (Figure 5B). A higher expression of Iba-1 was also observed in infected rats with respect to the control, but no differences were found between males and females (Figure 5B). Fluorescence intensity data are consistent with the number of cells expressing GFAP and Iba-1 (Figure 5C). All analyses were performed in samples from the left hemisphere, the region where parasites were detected by MRI and were proximal to the basal arachnoid (Figure 5D).

### 3.5. Depressed Proliferative Response in EXP-NCC

A decreased proliferative response of lymphocytes has been reported in chronic viral [29] and parasitic infections [30,31] due to anergy and cell exhaustion. A decrease in potential proliferative capacity induced by the non-specific activator concanavalin A was observed on day 150 post-infection (Figure 6). This decrease is likely due to lymphocyte dysfunction, as suggested by the low size and complexity gating. On the other hand, a negative correlation between HP-10 levels and proliferation (Figure 6C) supports the hypothesis that the decreased proliferation is due to cysticercal infection.

### 3.6. Behavior Assessment in Infected Rats

To evaluate any changes in rat behavior attributable to *T. crassiceps* infection, the horizontal and vertical movements of infected animals were recorded on day 120 post-infection and compared with sham and intact rats (controls). No differences were observed between control and infected groups in vertical movements in the central square. However, significant differences were found in horizontal movements (number of crossed lines) in control with respect to sham and infected rats (*p* < 0.05). With respect to exploration time in the central quadrant, no differences were found between infected rats and controls (Figure 7). Both movement types have been studied during hydrocephaly-induced damage, and vertical movements were more affected [32]. Our results suggest that no damage in motor functions was induced by the presence of parasites, which are still in a silent state [28], and the increase in horizontal movements could be due to stress.

## 4. Discussion

Experimental models of NCC have been developed in mice, rats, and pigs based on the use of *T. crassiceps* cysticerci [33], *Mesocestoides corti* cysticerci [34], and *T. solium* oncospheres [35]. However, all models available to evaluate cysticidal treatments include parenchymal and extraparenchymal cysts. On the other hand, studies based on these models have focused on the study of the inflammatory reaction induced by cysticidal treatment, and none has evaluated the effect of the anti-inflammatory treatments that usually accompany cysticidal therapy.

A promising murine model of EXP-NCC was reproduced in this study. Parasite establishment and its capacity to modulate the immune response were characterized. Various parameters that support its value as a model of the human disease were described, and they suggest that this experimental disease is an optimal model to evaluate new therapeutic approaches to improve the management of human EXP-NCC.

Interestingly, for the first time, it was demonstrated that the infection, as in humans, can be diagnosed by detecting the HP10 antigen in both CSF and serum, even though the latter showed lower values, as shown in Figure 2. This result is of particular interest as it suggests that, as in humans, a serological marker will allow us to follow the course of the infection.

As mentioned above, the treatment of EXP-NCC remains a challenge. On one hand, cysticidal drugs are only effective in less than 30% of cases [16]. On the other hand, the concomitant administration of glucocorticoids to prevent intracranial hypertension due to the exacerbated neuroinflammation following massive antigen release is imperative. However, there is evidence that glucocorticoids impair the efficacy of cysticidal treatment [19].

No symptoms attributable to this murine infection were observed, resembling the asymptomatic stage of human EXP-NCC, which may last for years, as circumstantially demonstrated in United Kingdom soldiers stationed in India, who were diagnosed with NCC years after their return [36]. This dilemma needs to be resolved to help symptomatic EXP-NCC patients.

In this regard, it is noteworthy that weak signs of inflammation were identified in our EXP-NCC model, as evidenced by a modest activation of astrocytes and microglia in female rats (Figure 5). Indeed, no cell infiltration was detected in regions proximal to established cysticerci in most rats (Figure 3). Similarly, neither microglial activation in the brain parenchyma nor increased CSF cellularity were observed, a scenario that resembles the clinical picture of human patients during the asymptomatic stage of NCC. Moreover, low CRP levels were measured in the serum of infected rats on day 180 post-infection, another finding that supports the weak inflammatory response observed. Further experiments are required to evaluate the levels of associated proinflammatory cytokines and how they affect the progression of the disease.

Another similarity between our model and EXP-NCC in humans is the decreased proliferative capacity of PBMCs. It should be noted that human EXP-NCC is accompanied by a depressed lymphocyte proliferative capacity [13]. This inhibition is associated with a central and peripheral increase in the number of Treg lymphocytes producing IL-10, the main regulatory cytokine known to be involved in this disease. A significant decrease in the levels of early- and late-activated lymphocytes is also observed in these patients [37,38]. Although we did not characterize the phenotype of the cells depressed in this model, it is feasible to assume that they are lymphocytes, considering the size and complexity of selected cells in the histograms (Figure 5). These results also resemble the depressed T cell responsiveness in intraperitoneal *T. crassiceps*-infected mice [39]. Considering this, it would be interesting to characterize these cells and their exhaustion status.

The use of MRI is indispensable to detect cysticerci in extraparenchymal locations. This tool has been optimized with the FIESTA sequence, which makes it possible to discriminate the signal emitted by CSF from the cysticercus itself [40]. In our study, cysticerci were detected by MRI on day 30 post-infection, as shown in Figure 3.

Another point that merits comment is the difference between this study and the one published by Hamamoto-Filho et al. [15]. No clear evidence of hydrocephalus was found in our animals, neither by MRI nor by histological analysis. Thus, further studies focused on the lateral ventricle should be conducted. This inability to detect hydrocephalus may be due to the smaller number of cysticerci implanted. Only 30 cysts per rat were used herein, compared with the 50 implanted in the previous study [15]. In line with this, rat behavior was not affected by the infection in our study. In fact, most of the behavior alterations were due to the surgery and not to the presence of extraparenchymal cysticerci; this trait resembles as well the asymptomatic stage of human EXP-NCC. OFT has been used extensively in murine models of NCC [28,32]. Other parasitic infections have been shown to alter exploratory activity in Wistar rats [41], so this test is useful in determining the general condition and stress of rats with EXP-NCC [42]. Although OFT can provide insight on the general locomotor activity in rats, further studies are needed to rule out other neuropathological, motor, and neurocognitive deficits, especially in cerebellum-related motor skills.

Overall, some parameters of the rat model of EXP-NCC that closely resemble the human disease are described in this study. The model is promising, and it will be used to evaluate new therapeutic approaches to control neuroinflammation, as well as new anti-cysticidal drugs and immunomodulatory treatments to restore and enhance specific anti-cysticercal immunity in EXP-NCC, with the goal of improving the management of this human disease.

## 5. Conclusions 

Herein, a rat model of EXP-NCC that closely resembles the human disease is described. This model will be employed to evaluate new therapeutic approaches to control neuroinflammation triggered by the cysticidal treatment and new therapeutic approaches to better manage this severe form of the disease.

## Figures and Tables

**Figure 1 brainsci-13-01021-f001:**
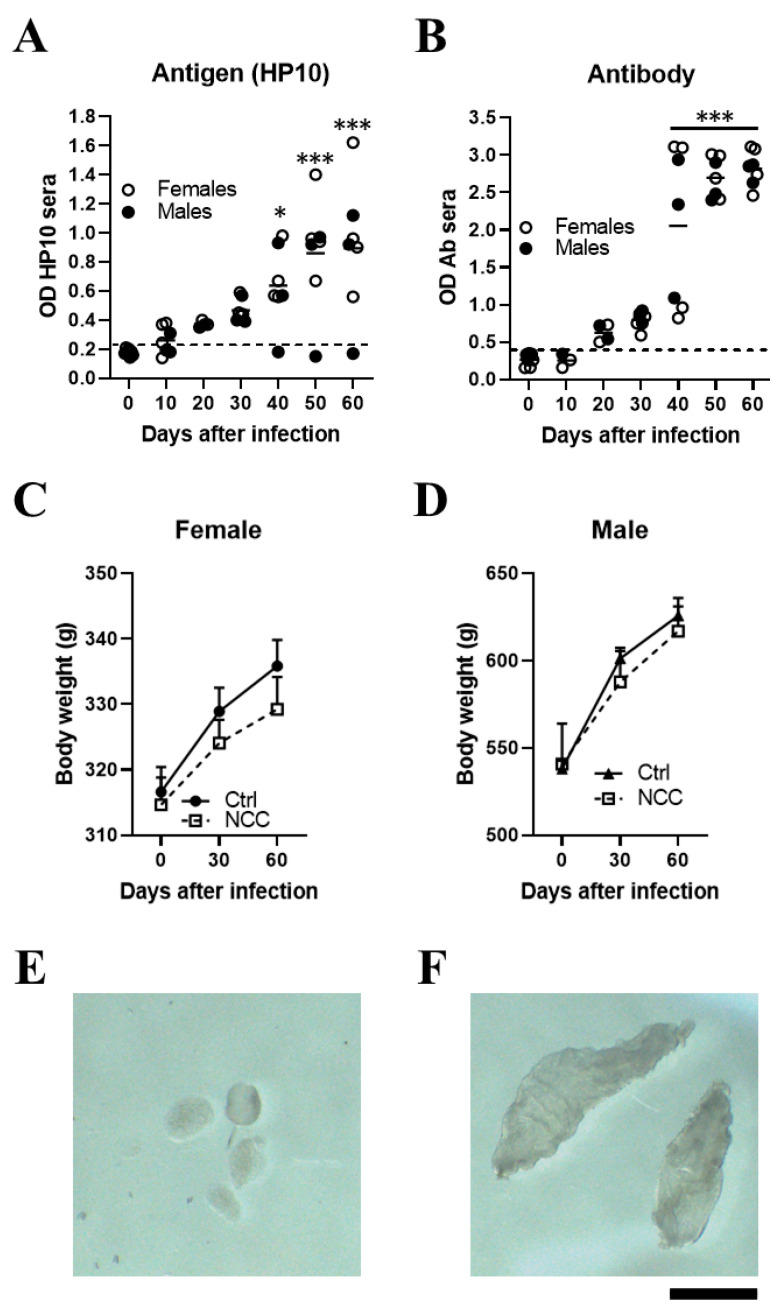
Inoculating *T. crassiceps* cysts into the cisterna magna of rats induced with neurocysticercosis. Levels of HP10 antigen (**A**) and anti-cysticercal antibodies (**B**) in the serum of rats experimentally infected with *T. crassiceps*. A time-dependent increase in antigen and antibody levels was observed. The cut-off value was calculated as the mean OD plus 3 SD in serum samples from seven NCC rats and the values on day 0. A sample was positive if the OD was higher than the cut-off value. Weight changes in infected female (**C**) and male (**D**) rats over a 60-day period. Mean parasite size before inoculation (400–500 µm) (**E**). A representative image showing the size of cysts recovered from CSF samples on day 60 post-infection (**F**) (*n*  =  7; 4 females and 3 males). Scale bar = 1 mm. Data are reported as mean ± SEM. Data normality was assessed with a Shapiro–Wilk test; all groups passed the test (*p* > 0.05). Antigen (F = 8.12; df = 6; *p* = 1.03 × 10^−5^) and antibody levels (F = 35.18; df = 6; *p* = 2.03 × 10^−13^) were compared by ANOVA, followed by the Dunnett test, using T0 as a control. *p* < 0.05 *; *p* < 0.001 ***.

**Figure 2 brainsci-13-01021-f002:**
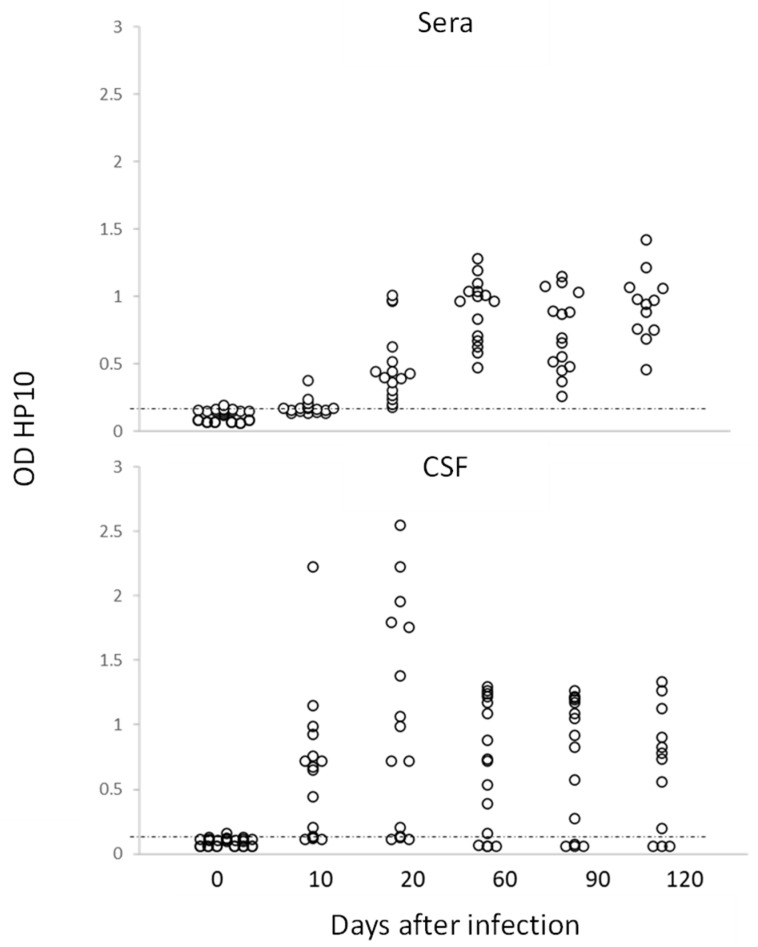
Levels of HP10 antigen in serum and CSF of rats over a 120-day period post-infection. The cut-off value was calculated as the mean OD plus 3 SD in CSF or serum samples from twenty non-NCC rat controls. A sample was positive if OD was higher than the cut-off value (*n*  =  20).

**Figure 3 brainsci-13-01021-f003:**
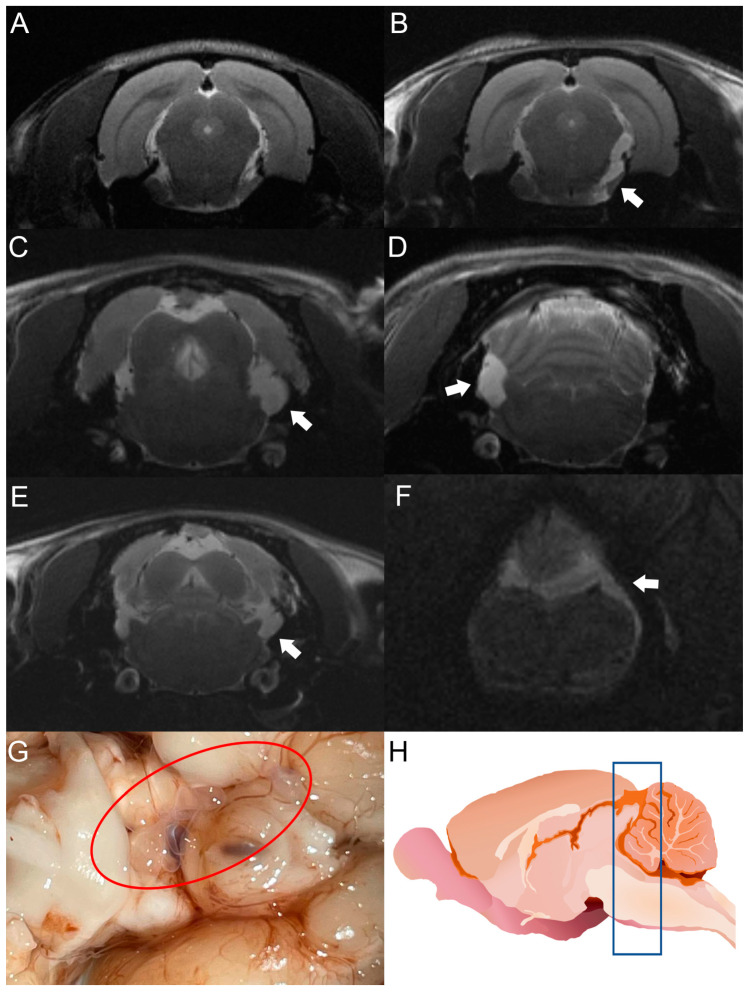
MRI studies of rats infected with *T. crassiceps* cysts at different times post-infection. Comparative parasite growth between day 30 (**A**) and day 120 (**B**) post-infection. (**C**–**F**) Photographs showing parasite growth in extraparenchymal regions of several animals on day 120 after infection. The location of the parasites is indicated by white arrows. (**G**) Macroscopic appearance of racemose cysticerci on day 60 post-infection (red circle). (**H**) Schematic representation showing the brain region analyzed by MRI (blue box). The brains of 5 infected animals were analyzed.

**Figure 4 brainsci-13-01021-f004:**
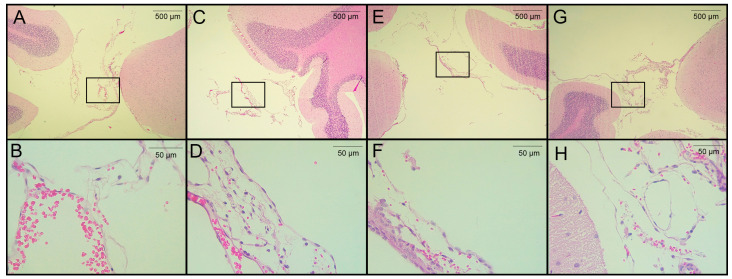
Histological analysis of H&E-stained brain sections from rats infected with *T. crassiceps* cysts on day 120 post-infection. (**A**,**B**) Coronal sections at different magnifications of the brain of a healthy rat in a region proximal to the cisterna magna (extraparenchymal/meningocortical region), with slight congestion and no obvious pathological change. (**C**) Region of the cisterna magna proximal to the cerebellum from an infected female rat. (**D**) Magnification of panel C, showing hemorrhagic meninges and edema unrelated to the presence of parasites. (**E**) Pericystic region in the cisterna manga of an infected female. (**F**) The magnified section shows few histopathologic changes. (**G**) Infected male, in the region of the cisterna magna proximal to the cerebellum (between the cerebellum and atlanto-occipital membrane) and magnification (**H**) with no remarkable histological changes. Images of 3 infected animals are shown. Scale bar = 500 and 50 µm. Black boxes show sites analyzed at higher magnification.

**Figure 5 brainsci-13-01021-f005:**
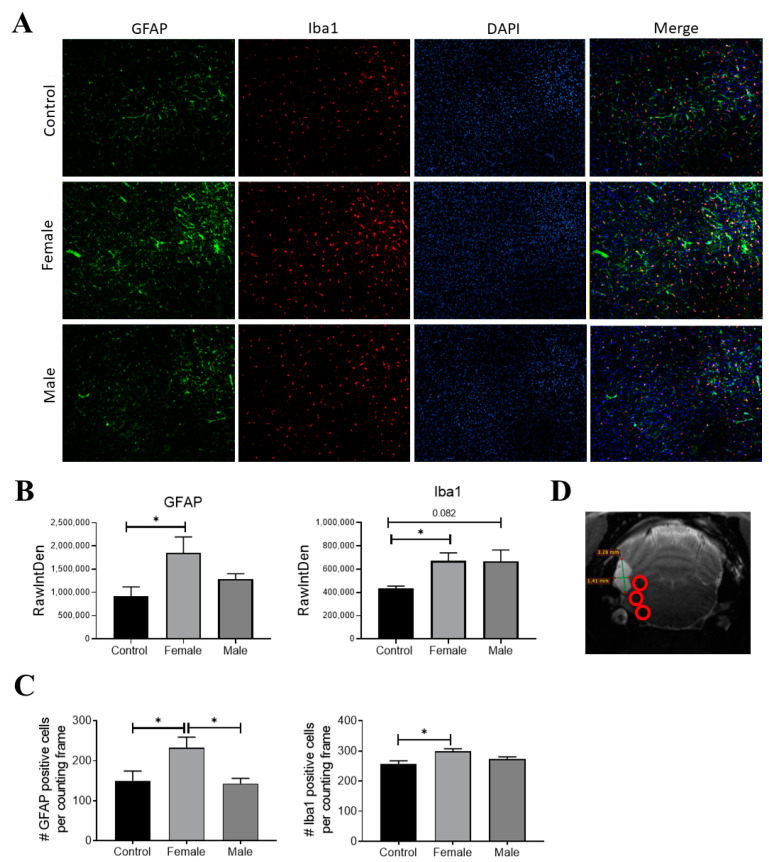
GFAP (green) and Iba-1 (red) expression in infected rats on day 120 post-infection. Nuclei were stained with DAPI (blue). (**A**) Representative rat brain sections from control (non-infected) and infected male and female rats. (**B**) Fluorescence intensity was quantified for GFAP and Iba-1 using the software Image J, expressed as RawIntDen (raw integrated density). (**C**) Number of cells that expressed GFAP and Iba-1. (**D**) Representative image of analyzed regions near cysticerci. Bar graph data report the mean of three adjacent brain slices, three areas of which, near the parasite locations (red circles), were analyzed. There are 9 photographs in total for each animal. Data normality was assessed with a Shapiro–Wilk test (*p* > 0.05). GFAP and Iba1 expression were reported as mean ± SEM. GFAP (F = 4.04; df = 2; *p* = 0.037) and Iba1 (F = 4.66; df = 2; *p* = 0.031) were compared with ANOVA–Tukey test. * Statistically significant differences, *p* < 0.05.

**Figure 6 brainsci-13-01021-f006:**
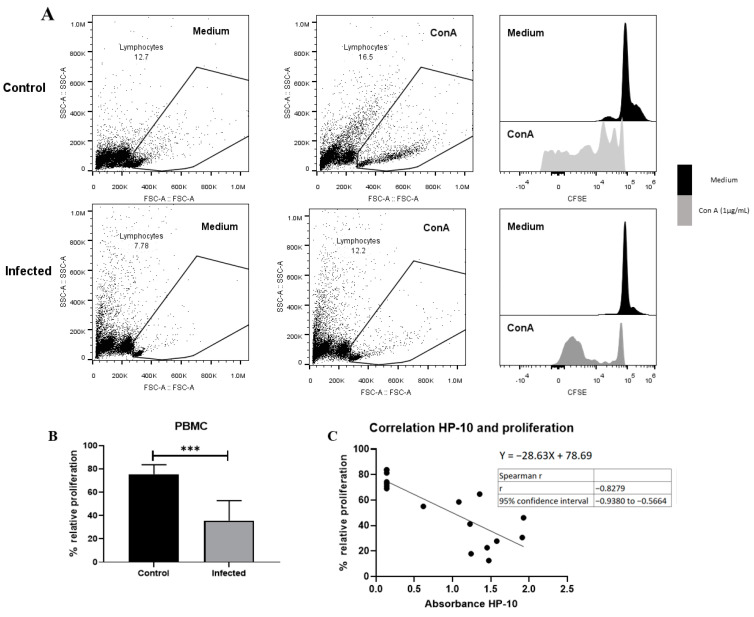
EXP-NCC by *T. crassiceps* cysticerci decreased PBMC proliferation in rats. (**A**) Representative dot plot and histograms of control and infected groups. A decrease in proliferative response was observed in cells with low complexity and small size (presumptive lymphocytes). (**B**) Infected rats showed lower relative proliferation than controls. Relative proliferation is shown as mean ± SD %. (**C**) Negative correlation between HP10 levels and PBMC proliferation. Data are shown as mean ± SEM (*n* = 8 per group). Data normality was assessed with a Shapiro–Wilk test (*p* > 0.05). Relative proliferation was compared with a two-tailed unpaired Student’s *t*-test (T = 5.643; df = 14; *p* = 5.76 × 10^−5^). Spearman correlation (rs = −0.82 and confidence interval −0.938 to −0.56, XYpairs: 17, *p* = 1 × 10^−4^) analysis was performed between relative proliferation and HP-10 levels. Statistically significant differences are labeled as *p* < 0.001 ***.

**Figure 7 brainsci-13-01021-f007:**
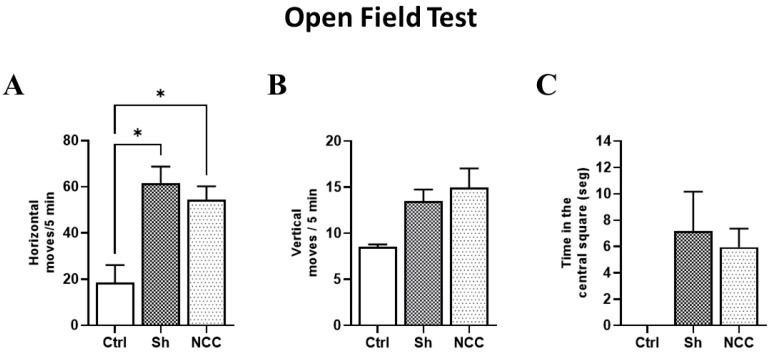
Behavior assessment in the open field test. Bar graphs of behavior parameters for control, sham, and infected rats. (**A**) Number of horizontal movements (transitions) during a 5 min period. (**B**) Vertical movements, defined as the number of times the rat stands on its hind legs during a 5 min period. (**C**) Time spent in the central square. * *p* < 0.05 vs. control. Data are shown as mean ± SEM (*n* = 4 for control and sham groups; *n* = 20 for NCC group, 10 females and 10 males). The normality of data on horizontal and vertical movements was assessed with a Shapiro–Wilk test, *p* > 0.05. Differences in horizontal and vertical movements were assessed with ANOVA, followed by the Tukey test: horizontal moves: F = 4.42, df = 2, and *p* = 0.0232; vertical moves: F = 1.249, df = 2, and *p* = 0.3047. Time in the central square has non-parametric distribution (Shapiro–Wilk test, *p* = 0.0079) and, thus, was compared with a Kruskal–Wallis test (KW, χ^2^ = 5.4; df = 2; *p* = 0.067). Statistically significant differences are labeled as *p* < 0.05 *.

## Data Availability

Data available in a publicly accessible repository. The data presented in this study are openly available in FigShare at https://doi.org/10.6084/m9.figshare.22799813.v1 accessed on 10 May 2023.

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
