# Peer review of "Standardizing an Experimental Murine Model of Extraparenchymal Neurocysticercosis That Immunologically Resembles Human Infection"

_brainsci, 2023, doi:10.3390/brainsci13071021_

Round 1
Reviewer 1 Report
No additional comments
Author Response
Thank you very much for reviewing our manuscript. After the review, we have improved the work.
Reviewer 2 Report
The manuscript "Murine extraparenchymal neurocysticercosis is accompanied by a weak immune-inflammatory response" by Espinosa-Ceron et al., shows interesting findings on a preclinical model of extraparenchymal neurocysticercosis (EXP-NCC) that are relevant to the field. Overal, experiments were well designed and presented, however, there are some issues that need to be cleared before the acceptance of the manuscript for publication.
- First of all, authors stated that inducing EXP-NCC in rats is accompanied by a weak immune-inflammatory response, according to the reduced proliferation of BBMC. However, it is on my concern whether pro-inflammatory cytokine levels are also less elevated in brain and/or blood, as have been observed in animal models of intracranially injected T. solium (doi: 10.1371/journal.pntd.0009295). By providing this data, authors can support the hypothesis of the manuscript.
- Why did the authors decided to do the final measures at 60-120 days? Some evidences using the intracranial inoculation of T-solium suggest that neuropatological, motor and neurocognitive deficits starts since 3 months to up to 1 year later (https://doi.org/10.3389/fncel.2023.1183322, doi: 10.1016/j.ajpath.2015.04.015).
- Measurement of the fluorescence of GFAP and IBA1 markers need to be contrasted with the number of IBA1 or GFAP cells and/or a wester blot of these markers.
- In the introduction section, authors described in detail the antiinflammatory treatment that patients with EXP-NCC receive, and also the poportion of patients that respond to this treatment. Can the authors expand in more detail, in the discussion section, on the responsibity to these treatments in animal models of NCC?
- Regarding with the results in the open field test. Authors suggested that the increase in horizontal movements in animals with EXP-NCC can be related with higher levels of stress. Can the authors provide a prove of this statement? (i.e. elevated-plus maze...). Also, as the inoculation and the growth of parasites is spatially related to the cerebellum, can the authors provide an assessment of cerebellar-related motor skills? (i.e. motor balance, cordination...).
- Regarding to the histologies, it is not mentioned the brain structure in which the measurements were performed.
Author Response
1. First of all, authors stated that inducing EXP-NCC in rats is accompanied by a weak immune-inflammatory response, according to the reduced proliferation of BBMC. However, it is on my concern whether pro-inflammatory cytokine levels are also less elevated in brain and/or blood, as have been observed in animal models of intracranially injected T. solium (doi: 10.1371/journal.pntd.0009295). By providing this data, authors can support the hypothesis of the manuscript.
R: As the reviewer mentioned, it would be crucial to measure the levels of cytokines during this infection. However, at this moment we do not have enough samples for this. Nevertheless, we have included additional data that support the occurrence of a weak inflammatory response in our revised version of the manuscript. The following sentence, summarizing these data, was added to the abstract in the revised manuscript:
“Low cell recruitment levels were observed surrounding established cysticerci in histological analysis, with a modest increase in GFAP and Iba1 expression in the parenchyma of female animals. Low cellularity in CSF and low levels of C reactive protein are consistent with a weak inflammatory response to this infection.” (Lines 33-36).
2. Why did the authors decided to do the final measures at 60-120 days? Some evidences using the intracranial inoculation of T-solium suggest that neuropatological, motor and neurocognitive deficits starts since 3 months to up to 1 year later (https://doi.org/10.3389/fncel.2023.1183322, doi: 10.1016/j.ajpath.2015.04.015).
R: As mentioned by the reviewer, it would certainly be interesting to evaluate neuropathological, motor, and neurocognitive deficits at longer times post-infection. However, in this study we optimized the experimental conditions to evaluate the effect of different treatments before the appearance of parasite-associated damage. Thus, for our purposes the absence of locomotor abnormalities was important, and a limit of 120 days-post infection was optimal for this.
3. Measurement of the fluorescence of GFAP and IBA1 markers need to be contrasted with the number of IBA1 or GFAP cells and/or a wester blot of these markers.
R: Considering the reviewer comment, the number of Iba-1 and GFAP positive cells was analyzed in all groups and described in the Material and Methods section. These results were included in the revised manuscript (Figure 5). As you can see, fluorescence intensity data are consistent with the number of cells that expressed those markers.
4. In the introduction section, authors described in detail the antiinflammatory treatment that patients with EXP-NCC receive, and also the proportion of patients that respond to this treatment. Can the authors expand in more detail, in the discussion section, on the responsibity to these treatments in animal models of NCC?
R: Considering the reviewer comment, a paragraph was added to the Discussion section of the revised manuscript:
“Experimental models of NCC have been developed in mice, rats, and pigs based on the use of T. crassiceps cysticerci [33], Mesocestoides corti cysticerci [34], and T. solium oncospheres [35]. However, all models available to evaluate cysticidal treatments include parenchymal and extraparenchymal cysts. On the other hand, studies based on these models have focused on the inflammatory reaction induced by cysticidal treatment, and none has evaluated the effect of the anti-inflammatory treatments that usually accompany cysticidal therapy.” (Lines 402-408).
5. Regarding with the results in the open field test. Authors suggested that the increase in horizontal movements in animals with EXP-NCC can be related with higher levels of stress. Can the authors provide a prove of this statement? (i.e. elevated-plus maze...). Also, as the inoculation and the growth of parasites is spatially related to the cerebellum, can the authors provide an assessment of cerebellar-related motor skills? (i.e. motor balance, cordination...).
R: Considering the reviewer comment, this sentence was edited. References were also added to support the use of OFT to evaluate stress in the rats. It would be of interest to evaluate in future experiments the tests that the reviewer mentioned. Thank you for your comments. This paragraph was added to the discussion section of the revised manuscript:
“OFT has been used extensively in murine models of NCC [28,32]. Other parasitic infections have been shown to alter exploratory activity in Wistar rats [41], so this test is useful in determining the general condition and stress of rats with EXP-NCC [42]. Although OFT can provide insight on the general locomotor activity in rats, further studies are needed to rule out other neuropathological, motor, and neurocognitive deficits, especially in cere-bellum-related motor skills.” (Lines 461-467).
6. Regarding to the histologies, it is not mentioned the brain structure in which the measurements were performed.
R: This information was added to the legend of Figure 4 and in the Results section of the revised manuscript:
“A histological analysis was performed on H&E-stained, paraffin-embedded sections from 4 of 7 rats. Cysticerci were found in 3 of the 4 animals studied. The parasites were located near the cisterna magna (between the cerebellum and the atlanto-occipital membrane), in the extraparenchymal/meningocortical region, or in the meninges near the brain stem.” (Lines 314-318)
Reviewer 3 Report
Firstly, I would like to congratulate the authors on their interesting and relevant work.
However, some points need to be addressed:
Introduction
1) It is advisable that the authors include the study design type within the title.
2) The authors should work with English editing services to improve the readability of the text.
3) I advise authors to include more information about the proliferative response of lymphocytes in NCC, inflammatory cytokines in NCC, neurocognitive alterations in NCC, and other relevant information about the pathogenesis related to any other tests they performed they find relevant in the introduction to provide more information to the reader on the rationale for the experiments, including their relationship with NCC. It is also interesting to provide details about NCC detection in different imaging modalities since they used MRI in their experiments.
4) In the abstract, line 20, what do the authors mean by: “ which in most cases does not compromise the patient's life”? Please rephrase as the sentence is not clear.
5) In line 22, I think it would be better phrased that “increased inflammation could lead to intracranial hypertension, and death in rare cases.”
6) In lines 39-40, please further specify “countries where living conditions promote its transmission”. Which countries? Which living conditions?
7) Please include more details about the global epidemiology including more percentages, and also provide further basic details about neurocysticercosis, including the larva cycle, the pathogenesis of the disease, and modes of transmission, risk factors in the introduction.
8) From lines 55-58, it would be interesting if the authors specified what is the approximate percentage of patients that have such rare complications and need to undergo those procedures.
9) From lines 54-56, it would be best if the authors rephrased “At some point in time, due to the longer residence time of the parasite or/and its detection by the host promoted by some concomitant circumstances, an exacerbated inflammatory process begins inducing symptoms that require emergency intervention.”. Which concomitant circumstances? What happens that leads to emergency intervention?
10) Please provide more details in the last paragraph of the introduction about the research question, scientific hypotheses of this research, and specific goals.
Methods
11) Was this entire study protocol created by the authors or was it similar to that of a previous study? If it was adapted from a previous study, the authors should briefly state what differed in their own study protocol.
12) Was this study protocol registered?
13) It is important to specify why the timeframes the authors used were chosen, if they were similar to previous studies, or if they had a rationale behind them. For example, why was the euthanasia 120 days after inoculation?
14) How was the sample size calculated?
15) Regarding the inclusion and exclusion of the animals for the experiments, please specify if they were established previously.
16) Were there any exclusions of animals in any of the analyses? If there were, please further specify why.
17) In each of the analyses that were done. please provide the exact number of animals in each experimental group.
18) Was randomization done to allocate experimental units to the control and treatment group? If it was, please provide randomization strategy used.
19) How did the authors decrease potential confounders in the experiments such as the order of inoculation of parasites or animal/cage location? If no strategies were used to decrease confounders, please explicitly state so.
20) Was there any blinding in the experiments?
21) The authors should specifically describe in the methods section which outcome measures were being assessed in the study, including primary and secondary outcome measures. A scientific rationale for the specific tests and experiments done in this study should be provided.
22) Please further specify which data were normal and which were not, as well as reliable statistical package used and its version. GraphPad PRISMA is generally used for the construction of graphs.
23) I suggest working with a statistician if the data distribution was not normal and trying to evaluate data distribution through frequency histograms because non-normality could violate the assumptions of ANOVA. There are several non-parametric tests that could substitute ANOVA in your case but a professional consultation would be more appropriate.
24) Please relevant information on the origin of the animals, health and immune status, genetic modification status, genotype, and any previous procedures.
25) Describe any interventions or steps taken in the experimental protocols to reduce pain, suffering and distress. Report expected or unexpected adverse events.
Results and Discussion
26) For every statistical analysis performed, the relevant parameters should be included. I suggest working with a statistician to improve statistical results reporting. For example, for the ANOVA results, F values, df, and significance should be reported in a table or in the text. For each experiment conducted, the authors should report: Summary/descriptive statistics for each experimental
group, with a measure of variability where applicable (e.g. mean and SD, or median and range), when available, the effect size with a confidence interval.
27) Please provide in the results and discussion a more in-depth analysis of the results, taking into account the study objectives and hypotheses, current theory and also adding more relevant studies in the literature, mainly in the discussion.
28) The authors should provide comments on the limitations of the study, including the above mentioned points, potential sources of bias, limitations of the animal model, and imprecision associated with the results.
29) Please provide a more in depth analysis of how the findings of this study are likely to generalise to other species or experimental conditions, including to humans.

Extensive editing required.
Author Response
Introduction
1) It is advisable that the authors include the study design type within the title.
R: The title is now modified in the new version considering the reviewer’s suggestion.
“Standardizing an experimental murine model of extraparenchymal neurocysticercosis that immunologically resembles human infection”
2) The authors should work with English editing services to improve the readability of the text.
R: The manuscript was revised by a native editor.
3) I advise authors to include more information about the proliferative response of lymphocytes in NCC, inflammatory cytokines in NCC, neurocognitive alterations in NCC, and other relevant information about the pathogenesis related to any other tests they performed they find relevant in the introduction to provide more information to the reader on the rationale for the experiments, including their relationship with NCC. It is also interesting to provide details about NCC detection in different imaging modalities since they used MRI in their experiments.
R: Considering the reviewer comments, the next paragraphs were added to the revised manuscript, as well as six references.
INTRODUCTION
“In these locations, the parasite can remain without causing symptoms for a long time [11], making diagnosis difficult, although magnetic resonance imaging (MRI) can help to assess the number, size, and location of parasites [12].” (Lines 71-73)
“…an exacerbated inflammatory process occurs, inducing intracranial hypertension and a depressed peripheral cellular immune response [13], with focal neurological deficits (16%) or recurrent seizures occurring in about 80% of symptomatic cases [14]; some degree of cognitive dysfunction has been reported in up to 88% of NCC patients, as well as significant deficits in motor control and impulsivity [14].” (Lines 75-80)
DISCUSION
“It should be noted that human EXP-NCC is accompanied by a depressed lymphocyte proliferative capacity [13]. This inhibition is associated with a central and peripheral increase in the number of Treg lymphocytes producing IL-10, the main regulatory cytokine known to be involved in this disease. A significant decrease in the levels of early and late activated lymphocytes is also observed in these patients [37,38].” (Lines 439-443)
4) In the abstract, line 20, what do the authors mean by: “which in most cases does not compromise the patient's life”? Please rephrase as the sentence is not clear.
R: This sentence was removed from the revised manuscript.
5) In line 22, I think it would be better phrased that “increased inflammation could lead to intracranial hypertension, and death in rare cases.”
R: Thank you, the sentence was edited as suggested.
6) In lines 39-40, please further specify “countries where living conditions promote its transmission”. Which countries? Which living conditions?
R: Additional information was included in the revised manuscript, as requested in points 6 and 7.
7) Please include more details about the global epidemiology including more percentages, and also provide further basic details about neurocysticercosis, including the larva cycle, the pathogenesis of the disease, and modes of transmission, risk factors in the introduction.
R: Several paragraphs were included in the Introduction section of the revised manuscript to provide further information, as requested in points 6 and 7.
“Cysticercosis, caused by the cystic larvae of the cestode Taenia solium, is an endemic disease in most countries of Latin America, sub-Saharan Africa, and Asia [1,2]. The main causes for the persistence of this parasitosis are lack of access to clean water and drainage, poor education, free-roaming pigs with access to contaminated human feces, and lack of control in pig trade and consumption [2]. The life cycle of the parasite includes an adult stage, the tapeworm that inhabits the human small intestine; this intestinal infection can be acquired by the ingestion of poorly cooked pork meat infected with cysticerci. Pigs be-came infected when they ingested eggs produced by this tapeworm in human feces. Hu-mans can also acquire cysticercosis by accidentally ingesting tapeworm eggs in contaminated food and/or water [2].
“The most serious form of the disease occurs when the parasite lodges in the central nervous system. Neurocysticercosis (NCC) is a foodborne neglected parasitic disease that causes 2.8 million disability-adjusted life years (DALYs) due to seizures, epilepsy, and intracranial hypertension [3].” (Lines 44-57).
8) From lines 55-58, it would be interesting if the authors specified what is the approximate percentage of patients that have such rare complications and need to undergo those procedures.
R: A paragraph including this information was added to the revised manuscript.
“… with focal neurological deficits (16%) or recurrent seizures occurring in about 80% of symptomatic cases [14]; some degree of cognitive dysfunction has been reported in up to 88% of NCC patients, as well as significant deficits in motor control and impulsivity [14].” (Lines 77-80).
9) From lines 54-56, it would be best if the authors rephrased “At some point in time, due to the longer residence time of the parasite or/and its detection by the host promoted by some concomitant circumstances, an exacerbated inflammatory process begins inducing symptoms that require emergency intervention.”. Which concomitant circumstances? What happens that leads to emergency intervention?
R: This paragraph was rephrased according to your suggestions:
“Eventually, due to the longer residence time of the parasite and/or its detection by the host immune system, or triggered by some concomitant inflammatory condition, an exacerbated inflammatory process occurs, inducing intracranial hypertension and a depressed peripheral cellular immune response [13]” (Lines 73-77).
10) Please provide more details in the last paragraph of the introduction about the research question, scientific hypotheses of this research, and specific goals.
R: This information was added at the end of the Introduction section in the revised manuscript.
“A rat model of EXP-NCC previously described by Hamamoto-Filho et al. (2015) [24] was reproduced with some modifications and characterized in our study with the aim of using it to evaluate new therapeutic approaches to restore the specific immunity against the parasite and more effective interventions to reduce neuroinflammation.” (Lines 104-107).
Methods
11) Was this entire study protocol created by the authors or was it similar to that of a previous study? If it was adapted from a previous study, the authors should briefly state what differed in their own study protocol.
This information was added to the revised manuscript:
“A previously described procedure [24] was used herein, with minor modifications in the number of parasites and the method for cysts injection. Briefly, …” (Lines 124-125).
12) Was this study protocol registered?
R: The protocol was registered and approved by our Institutional Committee for the Use and Care of Laboratory Animals, Instituto de Investigaciones Biomédicas, UNAM (approval number ID 6313).
This information was added to the Material and Methods section of the revised manuscript.
“All housing and experimental procedures were approved by the Institutional Committee for the Use and Care of Laboratory Animals of the Instituto de Investigaciones Biomédicas, UNAM, permit number ID 6313. Considering the stable conditions of the animals, no inclusion or exclusion criteria were established at the beginning of the study. Every effort was made, especially during surgery and euthanasia, to minimize animal suffering and stress.” (Lines 117-122).
13) It is important to specify why the timeframes the authors used were chosen, if they were similar to previous studies, or if they had a rationale behind them. For example, why was the euthanasia 120 days after inoculation?
R: This paragraph was added to the revised manuscript:
“The infection was monitored from day 0 to day 120 post-infection. On day 120 post-infection, the presence of cysticerci was confirmed by direct observation, microscopic analysis, HP10 detection, and MRI. No signs of distress or locomotor abnormalities were observed. The effect of different treatments was assessed at month 4 post-infection, before any damage associated with the presence of the parasite appeared.” (Lines 242-246).
14) How was the sample size calculated?
R: This information was added to the revised manuscript:
“The number of rats per group was estimated based on the results of Hamamoto-Filho et al. [28], considering the higher efficacy of infection in our study (60% versus > 90%).” (Lines 247-248).
15) Regarding the inclusion and exclusion of the animals for the experiments, please specify if they were established previously.
R: In this study, female or male Wistar rats (Rattus norvergicus) aged 8–9 weeks, bred in our pathogen-free vivarium, were employed. Considering the stable conditions of the animals, no inclusion or exclusion criteria were established at the beginning of the study. This information was added to the Material and Methods section of the revised manuscript:
“Considering the stable conditions of the animals, no inclusion or exclusion criteria were established at the beginning of the study.” (Lines 119-121).
16) Were there any exclusions of animals in any of the analyses? If there were, please further specify why.
R: This information was added to the Results section of the revised manuscript:
“HP10 levels remained low in one of the rats in the pilot experiment and no parasites were found on postmortem examination; thus, this animal was not included in immunofluorescence analysis.” (Lines 339-341).
17) In each of the analyses that were done. please provide the exact number of animals in each experimental group.
R: Details about the number of animals included in each experimental group were added in the corresponding figure captions.
18) Was randomization done to allocate experimental units to the control and treatment group? If it was, please provide randomization strategy used.
R: Each experiment was performed with a group of rats of the same age and sex, delivered in individual cages by our animal facilities. The animals were randomly selected as controls or infected.
This information was included in the revised manuscript:
“Groups of 7–10 female or male Wistar rats (Rattus norvergicus) aged 8–9 weeks were assigned to the treatments described below following a random variable simulation method implemented in R.” (Lines 110-112).
19) How did the authors decrease potential confounders in the experiments such as the order of inoculation of parasites or animal/cage location? If no strategies were used to decrease confounders, please explicitly state so.
R: Each rat was ear-tagged with a code with a consecutive number. This information was added to the revised manuscript. (Lines 114-115).
20) Was there any blinding in the experiments?
R: Some experiments were performed blind: the study of locomotor activity and lymphocyte proliferation experiments. This information was added to the revised manuscript (Line 237).
21) The authors should specifically describe in the methods section which outcome measures were being assessed in the study, including primary and secondary outcome measures. A scientific rationale for the specific tests and experiments done in this study should be provided.
As stated in the manuscript, infection success, as confirmed by macroscopic visualization of the parasite in the brain, histological analysis, and MRI, was the primary outcome. HP10 detection and the immunological profile and behavior that accompanied the infection are considered secondary outcomes.
The rationale of the tests and experiments is provided at the beginning of each method used in the revised manuscript.
22) Please further specify which data were normal and which were not, as well as reliable statistical package used and its version. GraphPad PRISMA is generally used for the construction of graphs.
R: Information on normality tests was added in the corresponding figure captions. On the other hand, all analyses were remade in another statistical package (R v.4.30). This information was added in the Statistical analysis section of the revised manuscript:
“Data normality was tested with the Shapiro-Wilk test. Differences between mean OD, be-havioral data (vertical and horizontal movements), body weight loss, and fluorescence in-tensity in immunofluorescence assays were evaluated by one-way ANOVA, followed by Dunnett or Tukey test. A two-tailed Student’s t-test was used to compare proliferation ca-pacity in PBMCs. Behavioral data in the central square were evaluated with a Krus-kal-Wallis test. Differences were considered as statistically significant when P-value was less than 0.05 *, 0.01 **, or 0.0001 ***. All analyses were carried out with GraphPad Prism® v.8.0 (GraphPad Software, San Diego, CA) and R v.4.30.” (Lines 249-256).
23) I suggest working with a statistician if the data distribution was not normal and trying to evaluate data distribution through frequency histograms because non-normality could violate the assumptions of ANOVA. There are several non-parametric tests that could substitute ANOVA in your case but a professional consultation would be more appropriate.
R: Normality tests were performed again. Non-parametric tests were applied when data failed to follow a normal distribution.
24) Please relevant information on the origin of the animals, health and immune status, genetic modification status, genotype, and any previous procedures.
R: All animals used in this study were exogamic rats, originally acquired from Harlan Laboratories and maintained and reproduced in our pathogen-free animal facilities under controlled temperature conditions, as stated in the Material and Methods section.
25) Describe any interventions or steps taken in the experimental protocols to reduce pain, suffering and distress. Report expected or unexpected adverse events.
R: As described in the Material and Methods section, all rats were anesthetized to perform implantation procedures, and when euthanized. No signs of pain or distress were detected.
Results and Discussion
26) For every statistical analysis performed, the relevant parameters should be included. I suggest working with a statistician to improve statistical results reporting. For example, for the ANOVA results, F values, df, and significance should be reported in a table or in the text. For each experiment conducted, the authors should report: Summary/descriptive statistics for each experimental group, with a measure of variability where applicable (e.g. mean and SD, or median and range), when available, the effect size with a confidence interval.
R: Additional information was included in figure captions, with the corresponding measure of variability, as well as the statistics used in each case.
27) Please provide in the results and discussion a more in-depth analysis of the results, taking into account the study objectives and hypotheses, current theory and also adding more relevant studies in the literature, mainly in the discussion.
R: Considering the reviewer’s comment, further details of the experiments were added to the Results section and expanded the Discussion section.
28) The authors should provide comments on the limitations of the study, including the above mentioned points, potential sources of bias, limitations of the animal model, and imprecision associated with the results.
R: The following paragraphs were added to the Discussion section in the revised manuscript:
“Further experiments are required to evaluate the levels of associated proinflammatory cytokines and how they affect the progression of the disease.” (Lines 435-437).
“Although OFT can provide insight on the general locomotor activity in rats, further studies are needed to rule out other neuropathological, motor, and neurocognitive deficits, especially in cerebellum-related motor skills.” (Lines 464-467).
29) Please provide a more in depth analysis of how the findings of this study are likely to generalise to other species or experimental conditions, including to humans.
R: A new paragraph was added at the end of the Discussion section.
“Overall, some parameters of the rat model of EXP-NCC that closely resemble the human disease are described in this study. The model is promising, and it will be used to evaluate new therapeutic approaches to control neuroinflammation, as well as immunomodulatory treatments to restore and enhance specific anti-cysticercal immunity in EXP-NCC, with the goal of improving our management of the human disease.” (Lines 468-472).
Round 2
Reviewer 2 Report
Thank you for consider all my comments to the manuscript.
Just as a suggestion, increase the quality of the images in figure 5. It is difficult to visualize the fluorescence, the images are too dark.
Author Response
1. Thank you for consider all my comments to the manuscript.
Just as a suggestion, increase the quality of the images in figure 5. It is difficult to visualize the fluorescence, the images are too dark.
R: Considering the reviewer's comment, we have increased the quality of the images in Figure 5. The new figure is now in the revised version.
Reviewer 3 Report
I would like to congratulate the authors since their manuscript has significantly improved after the first revision. My only request is regarding statistics that should be properly reported for all the analyses that were done. I will repeat the comment that should be addressed because I did not see the requested modifications in the figure legends.
26) For every statistical analysis performed, the relevant parameters should be included. I suggest working with a statistician to improve statistical results reporting. For example, for the ANOVA results, F values, df, and significance should be reported in a table or in the text. For each experiment conducted, the authors should report: Summary/descriptive statistics for each experimental group, with a measure of variability where applicable (e.g. mean and SD, or median and range), when available, the effect size with a confidence interval.
Minor editing required and typos should be removed.
Author Response
1. I would like to congratulate the authors since their manuscript has significantly improved after the first revision. My only request is regarding statistics that should be properly reported for all the analyses that were done. I will repeat the comment that should be addressed because I did not see the requested modifications in the figure legends.
For every statistical analysis performed, the relevant parameters should be included. I suggest working with a statistician to improve statistical results reporting. For example, for the ANOVA results, F values, df, and significance should be reported in a table or in the text. For each experiment conducted, the authors should report: Summary/descriptive statistics for each experimental group, with a measure of variability where applicable (e.g. mean and SD, or median and range), when available, the effect size with a confidence interval.
R: Thank you for all the comments and indications that have undoubtedly helped us substantially improve this work's presentation. This information is now in the revised version:
“Data are reported as mean ± SEM. Data normality was assessed with a Shapiro-Wilk test all groups pass the test (P > 0.05). Antigen (F= 8.12, df= 6, P= 1.03x10-5) and antibody levels (F= 35.18, df= 6, P= 2.03x10-13) were compared by ANOVA-Dunnett test, using T0 as a control. P < 0.05*; P < 0.01**; P < 0.001***)” (Lines 285-288).
“Data normality was assessed with a Shapiro-Wilk test (P > 0.05). GFAP and Iba1 expression were reported as Mean ± SEM. GFAP (F= 4.04, df= 2, P= 0.037) and Iba1 (F= 4.66, df= 2, P=0.031) were compared with ANOVA-Tukey test. *Statistically significant differences, P < 0.05.” (Lines 357-360).
“Data are shown as mean ± SEM (n = 8 per group). Data normality was assessed with a Shapiro-Wilk test (P > 0.05). Relative proliferation was compared with a two-tailed unpaired Student’s t-test (T= 5.643 df= 14, P=5.76x10-5). Spearman correlation (rs=-0.82 and confidence interval -0.938 to -0.56, XYpairs: 17, P=1x10-4) analysis was performed between relative proliferation and HP-10 levels. Statistically significant differences are labeled as P < 0.001***.” (Lines 376-380).
“Differences in horizontal and vertical movements were assessed with ANOVA-Tukey test; Hor-izontal moves (F= 4.42, df= 2, P= 0.0232), Vertical moves (F= 1.249, df= 2, P= 0.3047). Time in the central square has non-parametric distribution (Shapiro Wilk test, P=0.0079) so, was compared with a Kruskal-Wallis test (KW, χ2=5.4 df= 2, P= 0.067). Statistically significant differences are labeled as P < 0.05*.” (Lines 400-400).